# Correlation between Dental Composite Filler Percentage and Strength, Modulus, Shrinkage Stress, Translucency, Depth of Cure and Radiopacity

**DOI:** 10.3390/ma17163901

**Published:** 2024-08-06

**Authors:** Carolina Lopez, Bushra Nizami, Augusto Robles, Snigdha Gummadi, Nathaniel C. Lawson

**Affiliations:** 1Private Practice, Bogota 110110, Colombia; caalopezpe1@gmail.com; 2Division of Biomaterials, School of Dentistry, University of Alabama at Birmingham School of Dentistry, Birmingham, AL 35209, USA; nizami@uab.edu; 3Division of General Dental Sciences, School of Dentistry, University of Alabama at Birmingham School of Dentistry, Birmingham, AL 35209, USA; arobles@uab.edu (A.R.); sgummadi@uab.edu (S.G.)

**Keywords:** strength, modulus, shrinkage stress, translucency, depth of cure, radiopacity

## Abstract

Filler content in dental composites is credited for affecting its physical and mechanical properties. This study evaluated the correlation between the filler percentage and strength, modulus, shrinkage stress, depth of cure, translucency and radiopacity of commercially available high- and low-viscosity dental composites. Filler weight percentage (wt%) was determined through the burned ash technique (800 °C for 15 min). Three-point bend flexural strength and modulus were measured according to ISO 4049 with 2 mm × 2 mm × 25 mm bars. Shrinkage stress was evaluated using a universal testing machine in which composite was polymerized through two transparent acrylic rods 2 mm apart. Shrinkage was measured from the maximum force following 500 s. The translucency parameter (TP) was measured as the difference in color (ΔE00) of 1 mm thick specimens against white and black tiles. The depth of cure was measured according to ISO 4049 in a cylindrical metal mold (4 mm diameter) with a 10 s cure. Radiopacity was measured by taking a digital X-ray (70 kVp for 0.32 s at 400 mm distance) of 1 mm thick specimens and comparing the radiopacity to an aluminum step wedge using image analysis software. The correlation between the filler wt% and properties was measured by Pearson’s correlation coefficient using SPSS. There was a positive linear correlation between the filler wt% and modulus (r = 0.78, *p* < 0.01), flexural strength (r = 0.46, *p* < 0.01) and radiopacity (r = 0.36, *p* < 0.01) and negative correlation with translucency (r = −0.29, *p* < 0.01). Filler wt% best predicts the modulus and strength and, to a lesser extent, the radiopacity and translucency. All but two of the high- and low-viscosity composites from the same manufacturer had statistically equivalent strengths as each other; however, the high-viscosity materials almost always had a statistically higher modulus. For two of the flowable composites measured from the same manufacturer (3M and Dentsply), there was a lower shrinkage stress in the bulk-fill version of the material but not for the other two manufacturers (Ivoclar and Tokuyama). All flowable bulk-fill composites achieved a deeper depth of cure than the flowable composite from the same manufacturer other than Omnichroma Flow Bulk.

## 1. Introduction

In general, dental resin composites are composed of ceramic filler particles embedded within a matrix of a methacrylate resins. The filler composition and concentration in commercially available composites vary in order to achieve the desired mechanical, optical and handling properties of the material [1]. As the filler concentration is critical for characterizing a composite, many studies have measured the filler concentration of commercially available resin composites and correlated it to their mechanical and optical properties. 

The measurement of filler weight percentage (wt%) may be performed with thermogravimetric analysis in which the remaining weight of a sample of composite is recorded at increasing temperatures [2]. Alternatively, a burned ash method may be employed in which the weight of a sample of composite is measured following burning in a high temperature furnace [3,4,5]. Presumably, the actual wt% of a filler should be known by the manufacturer when the components of the composite were mixed during its fabrication. The wt% of composite measured by experimental means, however, does not always match that reported by the manufacturer [3,4]. 

Filler particles are ceramics, which are stronger and stiffer than resins, which are polymers. Therefore, fillers are credited for reinforcing the strength and increasing the modulus of resin composites. Although several previous studies have shown a positive correlation with filler wt% and modulus [2,3,5,6,7,8,9], the positive correlation between filler wt% and strength was often either weaker [2,6] or non-significant [7,8]. The decreased effect of filler weight on strength may be explained by the effect filler. Strength is affected by the initial fracture of composite at the filler/matrix interface, and stress concentrations are higher at the interface of large filler particles. Modulus, on the other hand, is determined by stress transfer between filler and matrix, which is not affected by filler size [10]. Also, filler particle agglomeration may affect strength as smaller nanoparticles may agglomerate at filler wt% above 40%, leading to decreased particle/matrix adhesion [11,12].

As only the resin component of resin composites is involved in polymerization, increasing the ratio of filler particles to resin content is credited with reducing shrinkage. Shrinkage can be measured as volumetric shrinkage by recording the change in the outline/volume of composite, shrinkage strain by measuring the dimensional change of an object containing a bonded composite, or shrinkage stress by measuring the force placed on a surface bonded to composite [13,14]. Volumetric shrinkage [15,16] and volumetric shrinkage strain [4,16,17,18,19] are both correlated with filler wt%. Shrinkage stress has also been shown to be correlated with filler wt% with experimental composites in which the filler wt% varied but the resin content was constant [20]. For commercially available composites, some found a correlation between filler wt% and shrinkage stress, whereas others did not [21,22]. Resin composition plays a larger role in determining shrinkage stress, whereas filler content has a larger effect on volumetric shrinkage [23]. Shrinkage stress rises with the conversion of the resin as well as the development of modulus in the composite. Increasing filler content reduces shrinkage but also increases modulus [24].

Filler particles within a composite act to scatter transmitted light through the material, and increasing the filler concentration will decrease light transmittance [25,26]. Larger filler particles lead to more light scattering and decreased transmittance [25,26]. The addition of small amounts of metal oxide opacifiers (<1%) can also have significant effects on the opacity of dental composites [27]. Resin content can also affect the translucency of filled composites with some methacrylate resins (Bisphenol A-diglycidyl dimethacrylate, BisGMA) appearing more translucent than others (Urethane dimethacrylate, UDMA and Triethylene glycol dimethacrylate, TEGDMA) [28]. Additionally, the match in the refractive index between the filler particles and resin will affect its translucency [29].

Filler particles may affect the depth of the cure of resin composites in many of the same ways that they affect the translucency of composites, by scattering transmitted light emitted from a curing unit. A difference is that the depth of cure is affected not only by the transmission of light through a resin composite but also the type and concentration of photoinitiators present in the composite. Additionally, the transmission of light through a composite during curing will change as the polymerization of the resin matrix will increase or decrease its match in the index of refraction with filler particles and, therefore, its ability to transmit light [30]. 

The radiopacity of a composite may be affected by filler content as any fillers containing barium, ytterbium, aluminum, or strontium may contribute radiopacity. Composites containing low-molecular-weight silica would not contribute significantly to radiopacity [31]. A previous study did not find a strong correlation between the filler wt% and radiopacity of commercially available composite [31]. Additionally, radiopacity is not correlated with the translucency of resin composites [32].

The objective of the study was to evaluate the correlation between the filler percentage and flexural strength, modulus, shrinkage stress, depth of cure, translucency and radiopacity of commercially available high- and low-viscosity dental composites. The novelty of this paper is that these correlations have not been previously performed on contemporary commercially available composites (including bulk-fill composites). The null hypothesis was that there would be no correlation between any of these properties. The secondary null hypothesis was that there would be no correlation between translucency and depth of cure or radiopacity. 

## 2. Materials and Methods

Sixteen composites were chosen for this study to represent high viscosity (packable and conventional) and low viscosity (flowable) materials as well as bulk-fill composites. Materials were chosen from different manufacturers to represent different resin compositions and filler types; however, those variables were not analyzed in this study. An A2 shade was used when available from the manufacturer and, if not, a universal shade was used. The materials are presented in Table 1.

### 2.1. Filler Weight Percentage

Samples (4 mm × 4 mm × 6 mm) of the composite (n = 8) were prepared in a mold and light-cured for 20 s (Elipar S10, 3M, St Paul, MN, USA; output = 1200 mW/cm^2^). Pilot testing revealed that light polymerization of samples did not affect filler weight percentage. Samples were weighed (W0) in an analytical balance with 0.0001 g accuracy (AE163, Mettler Toledo, Columbus, OH, USA). Samples were then placed into an alumina crucible (Coors high-alumina 20 mL crucible, Sigma Aldrich, St Louis, MO, USA), and the resin of each sample was burned out by heating the blocks in an electric furnace to 800 °C for 30 min. After 15 min of cooling, the remaining inorganic filler was then re-weighed (W1). The following formula was then used: filler wt% = (W1/W0 × 100%).

### 2.2. Flexural Strength Testing

The protocol from ISO 4049 [33] was used to measure flexural strength and elastic modulus of the materials. Bars (2 mm × 2 mm × 25 mm) of each composite material (n = 8) were prepared in split metal molds and light-cured (Elipar S10) for five 20 s overlapping cures on both sides. All specimens were lightly wet polished to 600 grit SiC paper. Specimens were stored in water for 24 h at 37 °C. Specimens were placed in a testing fixture with rod supports (2 mm diameter) that were 20 mm apart and had a 2 mm diameter loading indenter. The specimens were loaded at a rate of 1 mm/min until fracture in a universal testing machine (Instron 5565, Canton, MA, USA). Flexural strength was determined by the formula: 3 × maximum load × support span/2 × specimen width × (specimen height) [33].

Elastic modulus was determined by the internal software within the testing machine (Bluehill version 2, Instron).

### 2.3. Shrinkage Stress Testing

Two 6.5 mm diameter acrylic rods were cut to 18 mm lengths. One end of each rod (the bonding surfaces) was sanded with 600 grit sandpaper, primed with a 1:1 mixture of dichloromethane and methyl methacrylate, treated with Scotchbond Universal (3M), and light-cured for 20 s. The rods were secured into custom holders (Figure 1) in a universal testing machine (Instron 5565). The crosshead of the universal testing machine was positioned to allow a 2 mm distance between the cut ends of the rods. For high-viscosity composites, the distance between the rods was opened, composite was inserted between the rods, the 2 mm distance was then closed, and excess composite was removed by hand. For lowviscosity composite testing, a Teflon ring was placed around the circumference of the rods that contained two opposite-facing 0.75 mm diameter holes. Low viscosity composite was injected into one of the holes. The composites were cured (Elipar S10) for 20 s through the bottom rod. The 2 mm distance was maintained by the crosshead; however, compliance of the system was not adjusted for. The force applied to the load cell was recorded for 500 s. The maximum force applied was divided by the surface area of the rod to determine the shrinkage stress.

### 2.4. Translucency Testing

Discs (10 mm diameter × 1 mm thickness) of each composite material (n = 8) were prepared in molds and light-cured (Elipar S10) on both sides for three 20 s overlapping cures. Specimens were wet polished down to 1200 grit SiC on both sides in a polishing wheel. L*a*b* values were taken using a spectrophotometer (CM-700d; Konica Minolta, Ramsey, NJ, USA) against white-and-black calibrated tiles. A one-second sampling delay was used to minimize instrument vibrations during measurement. SCE (specular component excluded), SAV, and 10-degree geometry were used with each sample measured twice and then averaged together by the spectrophotometer. Translucency parameter was measured as the difference in color of the specimen measured against a white-and-black background using ΔE_00_ calculated with the CIEDE2000 color difference formula:$${∆E}_{00}={\left[{\left(\frac{{∆L}^{\prime }}{K_{L}S_{L}}\right)}^{2}+{\left(\frac{{∆C}^{\prime }}{K_{c}S_{C}}\right)}^{2}+{\left(\frac{{∆H}^{\prime }}{K_{H}S_{H}}\right)}^{2}+R_{T}\left(\frac{{∆C}^{\prime }}{K_{C}S_{c}}\right)\left(\frac{{∆H}^{\prime }}{K_{H}S_{H}}\right)\right]}^{1/2}$$
where ΔL′, ΔC′ and ΔH′ are lightness, chroma and hue differences; R_T_ (rotation function) accounts for the interaction between hue and chroma in the blue region; S_L_, S_C_ and S_H_ adjust for L*a*b* coordinate system variation; and K_L_, K_C_ and K_H_ are experimental condition corrections (all values of 1 for this study).

### 2.5. Depth of Cure Testing

Depth of cure of the materials was tested according to ISO 4049. Specimens were prepared by inserting composite material into a cylindrical metal mold with a 4 mm diameter. The length of the hole was 10 mm (ISO recommends 2 mm longer than the claimed depth of cure). The mold was placed on a mylar strip on a piece of white paper. A mylar film and a glass slide were placed over the composite with pressure until the composite was flush with the mold surface, and then the glass slide was removed. A curing light (Elipar S10) was placed against the glass in the center of the specimen and cured for 10 s. All materials were cured for 10 s for consistency. The mold was disassembled. The uncured composite was manually scraped away using a plastic spatula. The maximum length of the remaining specimen that was still intact (not removed by spatula) was measured with digital calipers. This length was divided in half to determine the depth of cure.

### 2.6. Radiopacity Testing

Discs (10 mm diameter × 1 mm thickness) of each composite material (n = 8) were prepared in molds and light-cured (Elipar S10) on both sides for three 20 s overlapping cures. Specimens were radiographed with a digital X-ray device (Planmeca; Prostyle Intra, Charlotte, NC, USA) set at 70 kVp for 0.32 s along with an aluminum step wedge at a distance of 400 mm. A standardized radiograph of the composites was made using an occlusal film 41.15 mm × 30.99 mm, and the processed film was scanned with a digital imaging system (Scan-X 1/0; Air Techniques Inc, Melville, NY, USA). Image analysis software (Photoshop version 22.0.1; Adobe Systems Inc., San Jose, CA, USA) was used to measure the gray scale levels of the composites using the histogram function and to compare intensity values to the step wedge.

### 2.7. Statistical Analysis

Data were checked for normality using Shapiro–Wilk’s test and variance homoscedasticity using Levene’s test, which used SPSS version 28.0.1.1 (IBM, Armonk, NY, USA). Statistical analyses were performed with a level of significance of α = 0.05. Filler wt%, flexural strength, modulus, shrinkage stress, translucency, depth of cure and radiopacity were each analyzed with a one-way ANOVA and Tukey post hoc analysis using SPSS software. A post hoc power analysis was conducted to determine the power of the tests at a significance using G*Power (version 3.1.9.6, Heinrich-Heine-Universität, Düsseldorf, Germany). Correlation between filler wt% and flexural strength, modulus, shrinkage stress, translucency, depth of cure and radiopacity, as well as between translucency and depth of cure and radiopacity, was evaluated with Pearson’s coefficient using SPSS.

## 3. Results

The mean ± standard deviation of the filler wt%, strength, modulus, shrinkage stress, translucency, depth of cure and radiopacity of all materials are presented in Table 2. The post hoc power analysis determined all tests had adequate power (>95%). A one-way ANOVA test determined significant differences between materials for filler wt%, strength, modulus, shrinkage stress, translucency, depth of cure and radiopacity (*p* < 0.001), with subsequent Tukey post hoc tests differentiating groups into statistically different rankings as indicated in Table 2.

The results of the Pearson coefficient to examine the correlation with filler weight revealed a statistically significant, strong positive correlation with modulus (r = 0.782, *p* < 0.001) and moderate positive correlation with flexural strength (r = 0.459, *p* < 0.001). This suggests that 61% of the variation in the modulus of different composites can be attributed to their filler percentage, and 21% of the variation in their strength can be attributed to their filler percentage. There was a statistically significant, weak positive correlation with radiopacity (r = 0.362, *p* < 0.001) and weak negative correlation with translucency (r = −0.286, *p* < 0.001). There was not a significant correlation between filler wt% and shrinkage stress (r = −0.138, *p* = 0.120) or depth of cure (r = −0.039, *p* = 0.664). The results of the Pearson coefficient to examine the correlation with translucency revealed a statistically significant, moderate positive correlation with translucency (r = 0.781, *p* < 0.001) and weak negative correlation with radiopacity (r = −0.355, *p* < 0.001). These relationships are demonstrated in Figure 2 and Figure 3.

## 4. Discussion

The objective of this study was to compare the filler wt% of commercially available composites to their mechanical and optical properties. Weight percentage was measured using the burned ash technique. The values of filler wt% were considerably lower than reported by their manufacturers (3–16% lower). The wt% reported by manufacturers would be determined by weighing the dry components of composite before adding them to the resin. Aside from filler, these dry components can also include opacifiers, pigments, initiators, radiopaques and resin in pre-polymerized fillers, as well as the silane coating on the fillers. Some of these components may have burned out before reaching 800 °C (for example, silanes burn out at 380–480 °C) [2]. The filler wt% reported in the current study is very similar to the values measured with the thermogravimetric analysis for Filtek Supreme Ultra (72% vs. 73%), GrandioSo (83% vs. 85%), Tetric Evoflow (71% vs. 70%) and GrandioSo Flow (77% vs. 78%) [2].

This study found correlations between filler wt% and strength, modulus, translucency and radiopacity; therefore, the first null hypothesis was partially rejected. The results of this study are similar to other studies, which have shown a strong correlation between filler wt% and modulus [2,3,5,6,7,8,9] and a weaker correlation with strength [2,6]. The strength of the composites are affected by fracture initiation at filler/matrix interfaces [10]. The lower correlation between filler wt% and strength is likely due to the many different types of fillers (irregular ceramic/glass particles, round glass particles, nanoclusters, silica and zirconia nanoparticles and pre-polymerized particles) and sizes used in the commercially available composites in this study. The values of strength in this study are slightly higher than reported in previous studies for the materials Filtek Supreme (85–120 MPa), GrandioSo (109–128 MPa), Tetric EvoCeram (77–92 MPa), Filtek Supreme Flow (95–97 MPa), GrandioSo Flow (83–110 MPa) and Tetric EvoFlow (98–118 MPa); however, the general ranking of materials is similar between this study and others [2,5,6,7,8]. The higher strength reported in this study could be due to the five overlapping cures on both sides from a high-power curing light (recommended by ISO) or the use of three-point bending rather than four-point bending tests. Of clinical interest, all but two of the high- and low-viscosity composites from the same manufacturer had a statically equivalent strength as each other; however, the high-viscosity materials almost always had a statistically higher modulus.

There was not a correlation between the filler wt% and shrinkage stress. These results could be explained by the larger role of the monomer in effecting shrinkage stress than filler load. Although most composites contained a mixture of BisGMA and UDMA, additional monomers were present in some materials. Additionally, there is a dual function of filler to decrease shrinkage but increase modulus [23,24]. A high modulus composite that shrinks the same distance as a low modulus composite will exert a higher shrinkage stress. A previous study reported a correlation between filler wt% and shrinkage stress [21]. In that study, four manufacturers were represented, and five of the composites tested were from the same manufacturer, which may have led to a homogeneity of the resin and filler composition. Additionally, the compliance of the stress-measuring device will dictate the effect of filler wt% on shrinkage stress. Compliance is a way to describe if there is any stress absorbance by the components of the stress-measuring device. If the device is compliant (for example, the grips connecting the clear rod to the load stretch when the composite begins contracting), stress measurements will be less affected by filler content and more by the shrinkage of the resin. If the device is less compliant, the increasing modulus caused by increased filler content will cause higher shrinkage stress [14]. Of clinical interest, for two of flowable composites measured within the same brand (3M and Dentsply), there was a lower shrinkage stress in the bulk-fill version of the material. For the other two brands (Ivoclar and Tokuyama), there was no difference in shrinkage stress in the bulk-fill version of the flowable composite.

The negative correlation between translucency and filler wt% has been reported previously [23,24]. The bulk-fill low-viscosity composites possessed lower translucency in their non-bulk-fill version (other than Tetric PowerFlow) despite having a higher filler weight percentage. The manufacturers of all bulk-fill flowable composites (other than Tokuyama) mention different filler types than their non-bulk-fill version. Therefore, the shape, size or match in the refractive index of these composites must have been adjusted to increase translucency. Tetric PowerFlow has been manufactured to have a match in the refractive index of resin and filler before cure but changes when the composite is cured. This change provides the clinical advantage of allowing curing light penetration but preventing unesthetic translucency after curing [30]. The Omnichroma composites all produced high translucency, which could be attributed to their relatively low filler percentage. These composites are formulated with 260 nm homogenous spherical fillers, which impart structural color. Structural color can be explained as the ability of the fillers to reflect back light in the yellow–red wavelengths, allowing coloration without pigments [34].

The depth of cure of the composites was not correlated with filler wt% but moderately correlated with translucency; therefore, the second null hypothesis can be rejected. The bulk-fill composites (aside from Omnichroma Flow Bulk) achieved a higher depth of cure, which may be related to their increased translucency. Tetric PowerFlow, however, was less translucent than its non-bulk-fill counterpart. This material achieved a deeper depth of cure due to the presence of Ivocerin, a more photactive initiator than camphorquinone [30]. The reason that the bulk-fill composite did not achieve the reported depth of cure in this study is that all composites were cured for the same amount of time (10 s) for consistency; however, the recommended curing time for the shades and curing light in this study would be greater for Filtek Bulk Fill Flow (20 s), SureFil SDR Flow + (25 s) and Omnichroma Bulk Fill Flow (30 s).

There was a weak positive correlation between radiopacity and filler wt% and negative correlation between radiopacity and translucency. Previous studies did not find significant correlations between radiopacity and translucency or filler wt% [31,32]. Radiopacity is likely affected by the presence of radiopaque additives. The manufacturer of the Omnichroma composites did not mention the addition of ytterbium fluoride on their scientific brochure, which may explain why these composites had the lowest radiopacity.

There are several limitations in this study. First, only a limited number of composites were tested, and the high-viscosity bulk-fill category of composites was not included in this analysis. There could also be improvements in the testing methodology, including measuring filler weight with thermogravimetric analysis, measuring strength with four-point bend testing, recording shrinkage stress beyond 500 s with a device that controls for compliance and measuring the depth of cure with hardness, degree of conversion or modulus development. In order to better predict composite properties through their characterization, future studies should consider an analysis of resin content. This analysis could involve measuring monomer composition, degree of conversion (over time) and glass transition temperature.

## 5. Conclusions

Within the limitation of the current study, the strongest correlations were noted between filler wt% and modulus as well as translucency and depth of cure. All but two of the high- and low-viscosity composites from the same manufacturer had statistically equivalent strengths as each other; however, the high-viscosity materials almost always had statistically higher modulus. For two of the flowable composites measured from the same manufacturer (3M and Dentsply), there was a lower shrinkage stress in the bulk-fill version of the material but not for the other two manufacturers (Ivoclar and Tokuyama). All flowable bulk-fill composites achieved a deeper depth of cure than the flowable composite from their same manufacturer other than Omnichroma Flow Bulk. The clinical implications of the depth of cure and shrinkage stress results demonstrate the advantages of bulk-fill flowable composites; however, the possible clinical disadvantage of a low modulus composite is not known.

## Figures and Tables

**Figure 1 materials-17-03901-f001:**
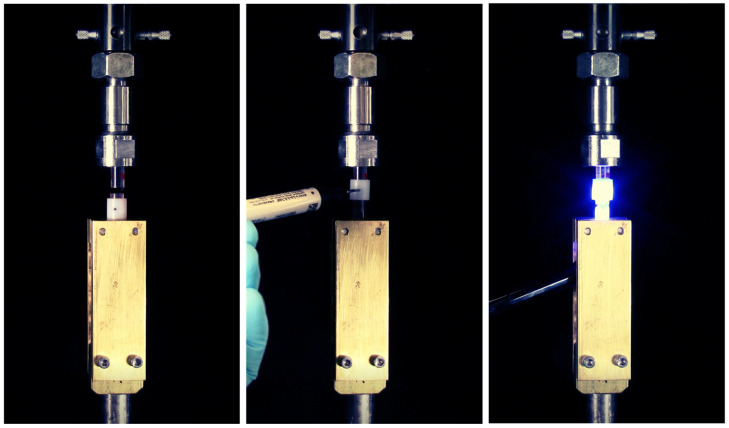
Shrinkage stress testing apparatus. (**left**) Two acrylic rods secured to load cell (upper) and base of testing machine (lower), (**center**) composite inserted through the 0.75 mm hole in Teflon ring, (**right**) composite cured through lower rod.

**Figure 2 materials-17-03901-f002:**
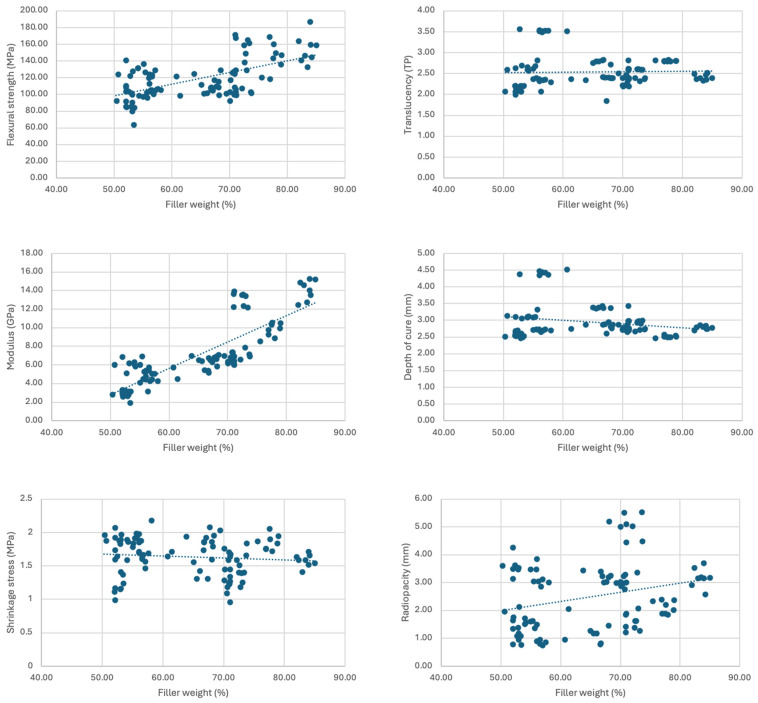
Filler weight plotted against flexural strength, modulus, shrinkage stress, depth of cure, translucency and radiopacity.

**Figure 3 materials-17-03901-f003:**
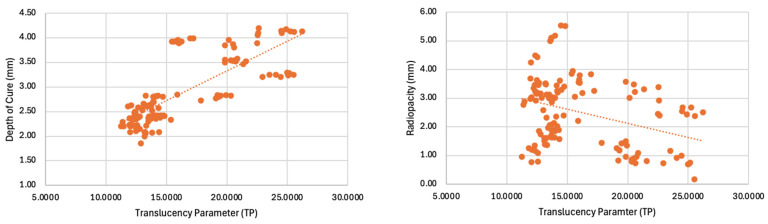
Translucency parameter plotted against depth of cure and radiopacity.

**Table 1 materials-17-03901-t001:** Materials used in this study.

Manufacturer	Material	Classification	* Filler wt%	* Filler Composition	* Monomer Composition
	Filtek Supreme Ultra	High viscosity	78.5%	20 nm silica, 4 to 11 nm zirconia and aggregated zirconia/silica cluster	BisGMA, UDMA, TEGDMA and BisEMA
3M St Paul, MN, USA	Filtek Supreme Flowable	Low viscosity	65%	20 nm silica, 75 nm silica, aggregated zirconia/silica cluster and ytterbium trifluoride (0.1 to 5.0 μm)	Procrylat, BisGMA and TEGDMA
	Filtek Bulk Fill Flowable	Low viscosity bulk-fill	64.5%	zirconia/silica (0.01 to 3.5 μm) ytterbium trifluoride (0.1 to 5.0 μm)	BisGMA, UDMA, BisEMA and Procrylat
Clinician’s Choice	Evanesce	High viscosity	82%	0.6 µm avg size glass fillers and silica	UDMA
London, ON, Canada	Evanesce Flow	Low viscosity	62%	0.5 µm avg size glass fillers and silica	UDMA
	TPH Spectra ST LV	High viscosity	75.5%	pre-polymerized barium glass spheres (15 µm), barium glass, methacrylic polysiloxane nanoparticles and ytterbium fluoride	UDMA, BisGMA and TEGDMA
Dentsply Sirona Charlotte, NC, USA	TPH Spectra ST Flow	Low viscosity	62.5%	pre-polymerized barium glass spheres (15 µm), barium glass and methacrylic polysiloxane nanoparticles, ytterbium fluoride	UDMA and BisGMA
	SDR Flow +	Low viscosity bulk-fill	70.5%	Barium glass, strontium alumino-fluoro-silicate glass, silica and ytterbium fluoride	Modified UDMA, TEGDMA and polymerizable dimethacrylate resin; polymerizable trimethacrylate resin
	Tetric Evoceram	High viscosity	75–76%	Barium glass, ytterbium fluoride, prepolymerized filler, spherical mixed oxide and silica	BisGMA and UDMA
Ivoclar Schaan, Liechtenstein	Tetric Evoflow	Low viscosity	60–65%	Barium glass, ytterbium fluoride, prepolymerized filler, spherical mixed oxide and silica	BisGMA, UDMA and D3MA
	Tetric PowerFlow	Low viscosity bulk-fill	71–72%	Barium glass, ytterbium fluoride, prepolymerized filler and silica	BisGMA, BisEMA, DCP and PCPEG-MA
	Omnichroma	High viscosity	79%	260 nm zirconia/silica	UDMA and TEGDMA
Tokuyama Tokyo, Japan	Omnichroma Flow	Low viscosity	70%	260 nm zirconia/silica	UDMA and 1,9-Nonanediol Dimethacrylate
	Omnichroma Flow Bulk	Low viscosity bulk-fill	69%	260 nm zirconia/silica	UDMA and TEGDMA
VOCO	GrandioSO	High viscosity	89%	1 µm glass, 20–40 nm silica	BisGMA, BisEMA and TEGMA
Cuxhaven, Germany	GrandioSO Flow	Low viscosity	81%	1 µm glass, 20–40 nm silica	HEDMA, BisGMA, BisEMA and TEGDMA

BisEMA = ethoxylated bisphenol-A dimethacrylate, D3MA = Dicandiol dimethacrylate, DCP = not stated, PCPEG-MA = not stated, HEDMA = hexanediylbismethacrylate. * Reported by manufacturer.

**Table 2 materials-17-03901-t002:** Mean ± standard deviation of properties testing in this study (materials in each column with different letters are statistically significantly different).

	* Filler wt%	Filler Weight (%)	Flexural Strength (MPa)	Modulus (GPa)	Shrinkage Stress (MPa)	Translucency (TP)	Depth of Cure (mm)	Radiopacity (mm)
Filtek Supreme Ultra	78.5%	72.0 ± 1.2 ^f^	158.8 ± 14.8 ^g^	13.2 ± 0.6 ^h^	1.3 ± 0.1 ^a,b,c^	13.1 ± 0.7 ^a,b,c^	2.6 ± 0 ^d^	1.8 ± 0.4 ^b,c,d^
Filtek Supreme Flowable	65%	53.8 ± 1.9 ^a,b^	127.6 ± 12.8 ^e,f^	6.3 ± 0.4 ^d,e,f^	1.9 ± 0.1 ^f,g^	13.7 ± 0.4 ^b,c^	2.6 ± 0 ^d^	1.7 ± 0.2 ^c,d^
Filtek Bulk Fill Flowable	64.5%	60.6 ± 1.8 ^c^	123.9 ± 20.8 ^d,e,f^	4.3 ± 0.2 ^b^	1.5 ± 0.1 ^c,d^	24.5 ± 1.6 ^f^	4.1 ± 0 ^i^	2.5 ± 0.3 ^e,f^
Evanesce	82%	70.3 ± 1.8 ^e,f^	115.6 ± 13 ^b,c,d,e^	6.7 ± 0.6 ^e,f^	1.2 ± 0.2 ^a,b^	12 ± 0.6 ^a^	2.2 ± 0.2 ^b^	3 ± 0.2 ^f,g^
Evanesce Flow	62%	52.7 ± 0.7 ^a^	82 ± 8.8 ^a^	2.8 ± 0.4 ^a^	1.2 ± 0.1 ^a^	12.3 ± 0.6 ^a,b^	2.2 ± 0 ^b^	1.2 ± 0.3 ^a,b^
TPH Spectra ST LV	75.5%	67.8 ± 2.4 ^d,e^	117.4 ± 11.8 ^b,c,d,e^	7.1 ± 0.5 ^f^	1.9 ± 0.1 ^g^	14.2 ± 0.3 ^c^	2.4 ± 0.1 ^c^	3.2 ± 0.2 ^g^
TPH Spectra ST Flow	62.5%	56.9 ± 2.6 ^b,c^	104.1 ± 9.2 ^a,b,c^	4.4 ± 0.2 ^b^	1.8 ± 0.2 ^e,f,g^	13.5 ± 1.1 ^b,c^	2.4 ± 0 ^c^	3.1 ± 0.6 ^f,g^
SDR Flow +	70.5%	67.5 ± 2.7 ^d,e^	108 ± 11 ^b,c,d,e^	4.8 ± 0.3 ^b,c^	1.5 ± 0.1 ^b,c,d^	20.6 ± 2 ^e^	3.9 ± 0.2 ^h^	3.2 ± 0.4 ^g^
Tetric Evoceram	75–76%	71.5 ± 2.1 ^f^	102.4 ± 2.7 ^a,b,c^	6.7 ± 0.5 ^e,f^	1.7 ± 0.1 ^d,e,f,g^	13.6 ± 1.1 ^b,c^	2.4 ± 0 ^c^	5 ± 0.5 ^h^
Tetric Evoflow	60–65%	52.8 ± 2.2 ^a^	101 ± 8.7 ^a,b,c^	3.1 ± 0.2 ^a^	1.8 ± 0.1 ^e,f,g^	13.2 ± 0.9 ^a,b,c^	2.1 ± 0 ^a^	3.5 ± 0.4 ^g^
Tetric PowerFlow	71–72%	67.6 ± 1 ^d,e^	100 ± 3.8 ^a,b^	5.8 ± 0.5 ^d,e^	1.7 ± 0.1 ^d,e,f,g^	16.1 ± 0.5 ^d^	3.9 ± 0 ^h^	3.6 ± 0.4 ^g^
Omnichroma	79%	65.6 ± 5.8 ^d^	105.4 ± 4.4 ^a,b,c,d^	5.9 ± 0.5 ^d,e^	1.6 ± 0.2 ^d,e,f^	19.3 ± 0.9 ^e^	2.8 ± 0 ^e^	1.2 ± 0.3 ^a,b,c^
Omnichroma Flow	70%	56.7 ± 2.9 ^b^	120.3 ± 6.7 ^b,c,d,e^	5.3 ± 0.3 ^c,d^	1.7 ± 0.1 ^d,e,f,g^	20.2 ± 0.9 ^e^	3.5 ± 0 ^g^	1 ± 0.2 ^a^
Omnichroma Flow Bulk	69%	53.7 ± 3.7 ^a,b^	124.2 ± 16 ^c,d,e,f^	5.4 ± 0.4 ^c,d^	1.8 ± 0.2 ^e,f,g^	24.5 ± 1 ^f^	3.3 ± 0 ^f^	0.8 ± 0.4 ^a^
GrandioSO	89%	83.5 ± 1 ^h^	160.4 ± 19.3 ^f,g^	14.3 ± 1.1 ^i^	1.6 ± 0.1 ^d,e^	12.5 ± 0.4 ^a,b^	2.4 ± 0 ^c^	3.2 ± 0.4 ^g^
GrandioSO Flow	81%	77.6 ± 1.3 ^g^	144.2 ± 15.7 ^g^	9.7 ± 0.7 ^g^	1.9 ± 0.1 ^f,g^	14.3 ± 0.9 ^c^	2 ± 0 ^e^	2.1 ± 0.3 ^d,e^

* Reported by manufacturer.

## Data Availability

All source data may be obtained from the corresponding author.

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
