# Peer review of "Correlation between Dental Composite Filler Percentage and Strength, Modulus, Shrinkage Stress, Translucency, Depth of Cure and Radiopacity"

_materials, 2024, doi:10.3390/ma17163901_

Round 1

Reviewer 1 Report

Comments and Suggestions for Authors

I have some observations:

1. Please add images more clearly and maybe you can replace with another type of diagram at Figure 2 and Figure 3 

2. The conclusion part must be improved with the obtained results.

3. At introduction part please add the novelty of the present study. Also, you can improve the introduction part with another studies in this field, see doi 10.1016/j.eurpolymj.2016.09.018; Human Dentine Remineralization Under Non-colagen Materials Action.

Author Response

  1. Please add images more clearly and maybe you can replace with another type of diagram at Figure 2 and Figure 3 

Thank you for this suggestion. I have improved the resolution of Figures 2 and 3 as well as increased their size. (page 8 and 9)

  1. The conclusion part must be improved with the obtained results.

The conclusions have been expanded to explain more of the results. (Page 11)

  1. At introduction part please add the novelty of the present study. Also, you can improve the introduction part with another studies in this field, see doi 10.1016/j.eurpolymj.2016.09.018; Human Dentine Remineralization Under Non-colagen Materials Action.

This is an excellent point. This statement has been added to the text: The novelty of this paper is that these correlations have not been previously performed on contemporary commercially available composites (including bulk-fill composites).” (Page 3)

Thank you for recommending the excellent paper.  The paper, however, describes a study to remineralize dentin whereas the current paper is addressing the mechanical properties of dental composites.  I am [of course] open to adding any additional references, however, I do not see how I could reasonably add this reference to the manuscript.

Reviewer 2 Report

Comments and Suggestions for Authors

Dear Authors, Very interesting work. With the development of dental materials science and the introduction of more and more new materials on the market, it is extremely important to test the properties of these materials. The study was precisely planned and executed.

From my function as a reviewer, I have a few minor comments:

- In Table 1, I would have added information on the resins and the type of filler in the tested materials and not only its weight percentage.

- "FIller weight percentage" Were the 4x4x6 mm samples fully polymerised after 20 seconds of polymerisation? Do you think incomplete polymerisation could have affected the results? 

- In Table 2 I would add a column with the Filler weight values given by the manufacturer.

- Please expand your conclusions. Please describe what the results tell clinicians. Should they use materials with higher or lower filler content. Or for which procedures which type of material to choose. 

Author Response

- In Table 1, I would have added information on the resins and the type of filler in the tested materials and not only its weight percentage.

This information has been added.  (Page 3 and 4 - Table 1)

- "FIller weight percentage" Were the 4x4x6 mm samples fully polymerised after 20 seconds of polymerisation? Do you think incomplete polymerisation could have affected the results? 

Thank you for bringing our attention to this detail. We had originally measured filler weight percentage with uncured specimens, but it was difficult to standardize the amount of composite. So we placed specimens in molds and cured them.  There was no different in the filler weight percentage of cured and uncured specimens.  I have added the following sentence: "Pilot testing revealed that light polymerization of samples did not affect filler weight percentage." (Page 4 - Filler weight percentage)

- In Table 2 I would add a column with the Filler weight values given by the manufacturer.

This has been added (Page 7 - Table 2)

- Please expand your conclusions. Please describe what the results tell clinicians. Should they use materials with higher or lower filler content. Or for which procedures which type of material to choose. 

The conclusions have been expanded and clinical relevance added. These are the specific sentences: "All but two of the high- and low-viscosity composites from the same manufacturer had statically equivalent strength as each other, however, the high-viscosity materials almost always had statistically higher modulus. For two of flowable composites measured from the same manufacturer (3M and Dentsply), there was a lower shrinkage stress in the bulk-fill version of the material but not for the other two manufacturers (Ivoclar and Tokuyama).  All flowable bulk-fill composites achieved a deeper depth of cure than the flowable composite from their same manufacturer other than Omnichroma Flow Bulk. The clinical implications of the depth of cure and shrinkage stress results demonstrate the advantages of bulk-fill flowable composites, however, the possible clinical disadvantage of a low modulus composite is not known."

(Page 11 - Conclusion)

Reviewer 3 Report

Comments and Suggestions for Authors

The paper entitled „Correlation between dental composite filler percentage and strength, modulus, shrinkage stress, translucency, depth of cure and radiopacity” focuses on evaluating the correlation between the filler percentage and flexural strength, modulus, shrinkage stress, depth of cure, translucency and radiopacity of commercially available high- and low-viscosity dental composites. Various characterization techniques have been used. The topic of the paper is interesting and relevant. The introduction is well-written, focusing on the issues addressed in the research. The results are nicely illustrated and well explained. However, comparing totally different dental composites can be misleading. In my opinion, the paper should be substantially improved before publishing because of the following main reasons:

1.               The abstract should be elaborated. Too many experimental details are present and few essential results are revealed.

2.               The citations in the text are not indicated appropriately. The numbers should be in square brackets.

3.               Some abbreviations such as BisGMA, UDMA, TEGDMA, should be explained.

4.               The main drawback of the study is that it compares very different types of composites whose properties depend on factors that are not accounted for in this study. In that way, the general conclusions made in the paper can vary substantially depending on the type and size of the filler, resin type and content, etc. It would be better, for the authors to indicate in Table 1, all details concerning the material composition (type and size of particles, resin type and content), not only its viscosity.

5.               When adding these details to Table 1, the discussion of the results should be improved taking into account the real composition of the dental composites.

6.               The conclusion section is too short. It should be improved by adding more information.

7.               The reference style should also be improved following the journal’s requirements.

Comments on the Quality of English Language

None

Author Response

  1. The abstract should be elaborated. Too many experimental details are present and few essential results are revealed.

We had attempted to keep the word count within those stated by journal .  But we have expanded the results as request.  (Page 1 - Abstract)

  1. The citations in the text are not indicated appropriately. The numbers should be in square brackets.

This has been corrected.

  1. Some abbreviations such as BisGMA, UDMA, TEGDMA, should be explained.

The full names of the molecules have been added.  This is the new sentence "Resin content can also affect the translucency of filled composites with some methacrylate resins (Bisphenol A-diglycidyl dimethacrylate, BisGMA) appearing more translucent than others (Urethane dimethacrylate, UDMA and Triethylene glycol dimethacrylate, TEGDMA) [29]." (Page 2 - second to last paragraph) 

  1. The main drawback of the study is that it compares very different types of composites whose properties depend on factors that are not accounted for in this study. In that way, the general conclusions made in the paper can vary substantially depending on the type and size of the filler, resin type and content, etc. It would be better, for the authors to indicate in Table 1, all details concerning the material composition (type and size of particles, resin type and content), not only its viscosity.

We have added all the information about the composites we can find from the manufacturers scientific brochures to Table 1. (Page 3 and 4 - Table 1).

  1. When adding these details to Table 1, the discussion of the results should be improved taking into account the real composition of the dental composites.

Thank you for this suggestion.  We were able to use information from Table 1 to better explain results in the Discussion. The following sentences have been added or modified: 

The lower correlation between filler wt% and strength is likely due to the many different types of fillers (irregular ceramic/glass particles, round glass particles, nanoclusters, silica and zirconia nanoparticles, pre-polymerized particles, pre-polymerized barium glass spheres) and sizes used in the commercially available composites in this study.  (Page 9 - 2nd paragraph)

Although most composites contained a mixture of BisGMA and UDMA, additional monomers were present in some materials. (Page 9 - 3rd paragraph)

The manufacturers of all bulk-fill flowable composites (other than Tokuyama) mention different filler types than their non-bulk fill version. (Page 10 - 2nd paragraph)

The manufacturer of Omnichroma composites did not mention the addition of ytterbium fluoride on their scientific brochure which may explain why these composites had the lowest radiopacity. (Page 10 - 4th paragraph)

  1. The conclusion section is too short. It should be improved by adding more information.

The conclusions have been expanded and clinical relevance added. These are the specific sentences: "All but two of the high- and low-viscosity composites from the same manufacturer had statically equivalent strength as each other, however, the high-viscosity materials almost always had statistically higher modulus. For two of flowable composites measured from the same manufacturer (3M and Dentsply), there was a lower shrinkage stress in the bulk-fill version of the material but not for the other two manufacturers (Ivoclar and Tokuyama).  All flowable bulk-fill composites achieved a deeper depth of cure than the flowable composite from their same manufacturer other than Omnichroma Flow Bulk. The clinical implications of the depth of cure and shrinkage stress results demonstrate the advantages of bulk-fill flowable composites, however, the possible clinical disadvantage of a low modulus composite is not known."

(Page 11 - Conclusion)

  1. The reference style should also be improved following the journal’s requirements.

This has been corrected.  (Pages 11 and 12 - References)

Round 2

Reviewer 1 Report

Comments and Suggestions for Authors

I accept for publication

Reviewer 3 Report

Comments and Suggestions for Authors

The authors have carefully addressed the reviewer's recommendations.